# 3D reconstructions of parasite development and the intracellular niche of the microsporidian pathogen *Encephalitozoon intestinalis*

Noelle V. Antao[1], Cherry Lam[1], Ari Davydov[1], Margot Riggi[2], Joseph Sall[3], Christopher Petzold[3], Feng-Xia Liang [1,3], Janet H. Iwasa[2], Damian C. Ekiert [1,4] ✉ & Gira Bhabha [1] ✉

Microsporidia are an early-diverging group of fungal pathogens with a wide host range. Several microsporidian species cause opportunistic infections in humans that can be fatal. As obligate intracellular parasites with highly reduced genomes, microsporidia are dependent on host metabolites for successful replication and development. Our knowledge of microsporidian intracellular development remains rudimentary, and our understanding of the intracellular niche occupied by microsporidia has relied on 2D TEM images and light microscopy. Here, we use serial block-face scanning electron microscopy (SBF-SEM) to capture 3D snapshots of the human-infecting species, *Encephalitozoon intestinalis*, within host cells. We track *E. intestinalis* development through its life cycle, which allows us to propose a model for how its infection organelle, the polar tube, is assembled de novo in developing spores. 3D reconstructions of parasite-infected cells provide insights into the physical interactions between host cell organelles and parasitophorous vacuoles, which contain the developing parasites. The host cell mitochondrial network is substantially remodeled during *E. intestinalis* infection, leading to mitochondrial fragmentation. SBF-SEM analysis shows changes in mitochondrial morphology in infected cells, and live-cell imaging provides insights into mitochondrial dynamics during infection. Our data provide insights into parasite development, polar tube assembly, and microsporidia-induced host mitochondria remodeling.

Microsporidia, related to fungi, are obligate intracellular pathogens[1] that infect a wide range of invertebrate and vertebrate hosts, including humans[2–9]. In humans, microsporidia most often infect the gastrointestinal tract and cause diarrheal diseases that are self-limiting, but in immunocompromised patients, infections can be fatal[5,10–12]. Microsporidia infect ~5% of the human population[13] and have significant impacts on agriculture, aquaculture, and other industries[14]. Despite their prevalence and global impact, microsporidian biology

[1]Department of Cell Biology, New York University School of Medicine, New York, NY, USA. [2]Department of Biochemistry, University of Utah, Salt Lake City, USA. [3]Office of Science and Research Microscopy Laboratory, New York University School of Medicine, New York, NY, USA. [4]Department of Microbiology, New York University School of Medicine, New York, NY, USA. ✉e-mail: damian.ekiert@ekiertlab.org; gira.bhabha@gmail.com

remains understudied. The species *Encephalitozoon intestinalis*[5] is one of the most common causes of microsporidian infection in humans. Like other microsporidian species, *E. intestinalis* employs a unique, harpoon-like invasion organelle called the polar tube (PT) to gain entry into a new host cell[15–17]. The PT, many times the length of a microsporidian cell, is coiled up inside the dormant spore. To initiate infection, spores undergo germination. During germination, the PT rapidly extends to form a long, linear tube, through which infectious cargo, called sporoplasm, is transported from the spore to the host cell[18–21] (Fig. 1A). Once inside the host cell, *E. intestinalis* replicates within a parasitophorous vacuole (PV)[5,22,23], also known as the sporophorous vesicle[24]. Similar to *E. intestinalis*, many other microsporidian species replicate within PVs, while other species replicate directly in the host cytosol[25,26]. The parasite replicates through a series of 4 stages: (1) the meront stage, an amorphous, multinucleate stage in which repeated nuclear divisions occur without cytokinesis[5,22,23]; (2) the sporont stage, in which cellularization occurs and the spore coat begins to form[5,22,23]; (3) the sporoblast stage, a post-replicative stage in which parasites start developing the specialized organelles found in mature spores, such as the PT[5,22,23]; and (4) the spore stage, which is the transmissible form of the parasite[5,22,23]. Spores are surrounded by a thick spore coat, which consists of a proteinaceous exospore layer, and a polysaccharide-rich endospore layer that contains chitin. The spore coat renders the spores resistant to the environment.

Microsporidia are highly diverged from other eukaryotes. As an obligate intracellular parasite with the smallest known eukaryotic genome[27], *E. intestinalis* lacks numerous pathways important for basic cellular metabolism, such as the electron transport chain[27,28], and is highly dependent on the host for resources such as metabolites. For these reasons, microsporidia must establish an intracellular niche in order to avoid host detection and to promote its own replication, development and dissemination. Establishing an intracellular niche likely involves modulation of host cell physiology, in order to:

(1) optimally position the PV within the host cell, (2) expand the PV membrane to accommodate the replicating parasites, and (3) siphon metabolites into the PV. Evidence from some *Encephalitozoon spp.* suggests that the PVs are in close proximity to the host mitochondrial network and nucleus[29–31], which may facilitate the acquisition of ATP, lipids, and other molecules from the host. Interactions between mitochondria and PVs have been observed using 2D TEM as well as light microscopy, and a direct interaction between microsporidia and mitochondrial outer membrane proteins has also been proposed[29]. More recent data suggest that infection with some *Encephalitozoon spp.* can lead to Drp1-dependent mitochondrial fragmentation[32].

The physical basis of the interactions within the *E. intestinalis* niche, and how the niche is established remain unclear. Moreover, our knowledge of how microsporidian parasites develop in a PV, and how organelles like the PT are built and organized during development remains rudimentary. To learn about these processes, 3-dimensional information on different parasite stages, and interactions between the parasite and the host in infected cells are needed. Advances in volume EM (vEM) techniques have enabled the reconstruction of large biological samples in 3D, at nanometer resolution, through a depth of several micrometers[33]. This makes vEM techniques, such as serial block-face scanning electron microscopy (SBF-SEM)[34], excellent tools to study parasites and their interactions with hosts. vEM approaches have recently been used to investigate the ultrastructure of microsporidia and other eukaryotic parasites, both in isolation[16] and during intracellular development[35,36]. To gain a better understanding of the *E. intestinalis* life cycle, intracellular parasite development, and host-cell interaction, we used SBF-SEM, as well as live-cell imaging. Our work reveals how parasite organelles develop throughout the *E. intestinalis* life cycle, the physical connections between the PV and host organelles, and the dynamics of parasite-mediated host mitochondrial remodeling. Using these data, we propose a model for how the PT and other organelles develop within the parasite.

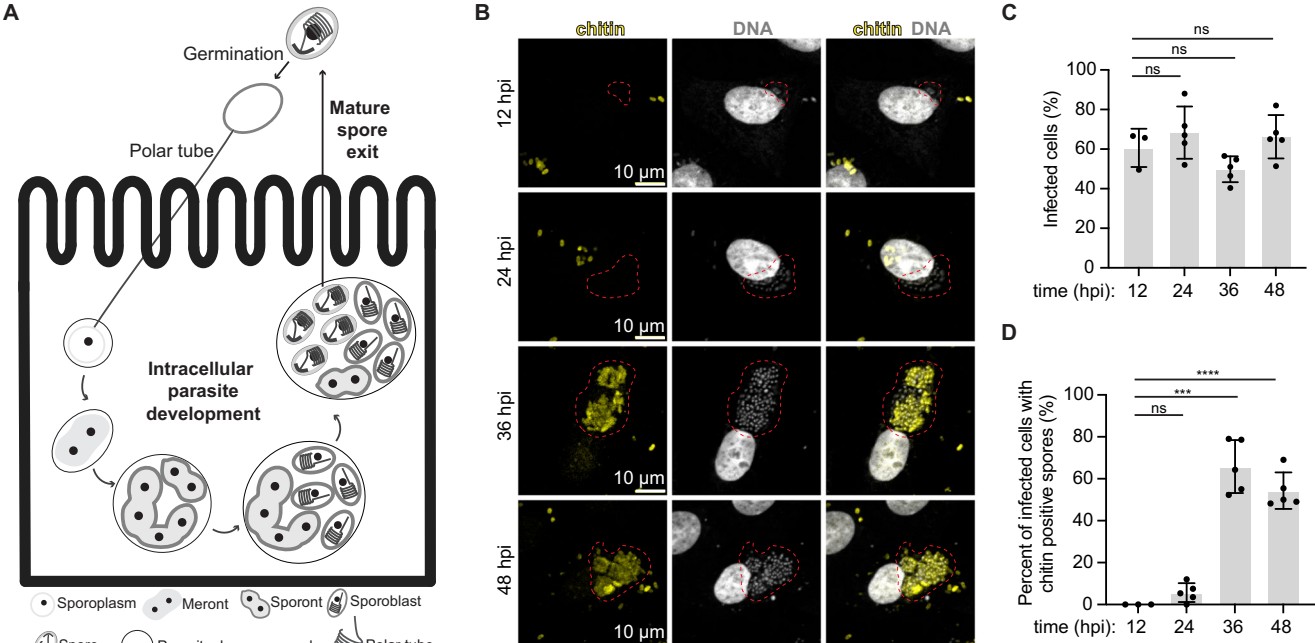

**Fig. 1 | Monitoring *E. intestinalis* life cycle, intracellular parasite development and infectivity in Vero cells. A** Schematic of the *E. intestinalis* life cycle. **B** Micrographs of Vero cells infected with *E. intestinalis* and imaged at 12 hpi, 24 hpi, 36 hpi and 48 hpi. Cells are stained for DNA (DRAQ5, gray), chitin (calcofluor white, yellow). Infection clusters in cells are marked with a red dotted line. **C** Quantification of infected Vero cells at 12 hpi, 24 hpi, 36 hpi and 48 hpi.

**D** Quantification of infected Vero cells that have chitin-positive infection clusters at 12 hpi, 24 hpi, 36 hpi and 48 hpi. Mean ± SD are from three biological replicates for 12 hpi and five biological replicates for 24 hpi, 36 hpi and 48 hpi. $n = 100$ cells per experiment. ****$p < 0.0001$, ***$p < 0.001$, ns, not significant ($p > 0.05$), calculated from an unpaired Student's *t*-test for (**C**, **D**). Source data for (**C** and **D**) are provided in the Source Data file.

## Results

### Correlative light and electron microscopy (CLEM) of host cells infected with *E. intestinalis*

To better understand the intracellular development and the host cell niche, we set out to obtain 3D snapshots of *E. intestinalis*-infected host cells using SBF-SEM[34]. To establish a timeline for the *E. intestinalis* life cycle in Vero cells and identify optimal infection time points for SBF-SEM experiments, we monitored infection using light microscopy. We used a fluorescent DNA dye (DRAQ5)[37,38], which stains parasites of all developmental stages, and a fluorescent chitin dye (calcofluor white)[39], which stains the spore coat, to detect parasites at more mature developmental stages. Meronts, sporonts and sporoblasts are expected to be DRAQ5-positive only, while spores, which have developed a chitin-rich endospore layer, are expected to be DRAQ5-positive and calcofluor white-positive. At all the time points analyzed, at least 50% of cells were infected (Fig. 1B, C). Very few chitin-positive infected cells are observed at early time points (12 hpi and 24 hpi), while more than 55% of infected cells are chitin positive at later time points (36 hpi and 48 hpi) (Fig. 1D). These data suggest that 12–24 hpi time points provide information on early parasite development, while 36–48 hpi time points provide information on more mature parasites. We therefore decided to image infected cells at 24 hpi and 48 hpi using SBF-SEM.

To prepare samples for SBF-SEM, we used a correlative light and volume EM (CLEM)[40] workflow that we optimized for *E. intestinalis*-infected Vero cells (see "Methods"). The CLEM workflow was necessary in order to efficiently identify and image infected cells amidst neighboring uninfected cells. We first seeded Vero cells on a tissue culture dish with a gridded coverslip that has an identifiable pattern within each square. Following infection with *E. intestinalis* spores for either 24 h or 48 h, we used DIC light microscopy to identify infected cells of interest and record their location on the gridded coverslip. After fixation and embedding for EM, we used the grid location to relocate cells of interest as identified by light microscopy and examined them at an ultrastructural level using SBF-SEM (Fig. 2A, B). For each infection time point, we acquired two datasets, with each dataset containing a single infected cell (Fig. 2C, D; Supplementary Fig. 1A, C). In addition, we acquired three datasets for subcellular regions of infected cells at 48 hpi, with each region containing at least one parasitophorous vacuole (PV).

Across the seven datasets, we observed a total of 17 PVs: 11 PVs at 24 hpi and 6 PVs at 48 hpi (Supplementary Data 1). Overall, the PVs observed in Vero cells were flat, pancake-like structures (Fig. 2E, F; Supplementary Fig. 1B), consistent with the Vero cells themselves being relatively flat. We annotated, segmented and reconstructed PVs and parasites in 3D, to assess how parasites are packed within the PV. Consistent with previous TEM data, parasite development is not synchronous, and a single PV can contain parasites at various developmental stages[5,22,23]. As expected from our observations by light microscopy, in PVs at 24 hpi, predominantly earlier stages of parasite development are present. These are sporonts (60%) and sporoblasts (40%), which both lack a clear endospore layer (Fig. 2C, E, F; Supplementary Fig. 1A, B; Supplementary Fig. 2A). Although a proliferative, multinucleate plasmodial stage, or meront, is common in many microsporidian species, in *E. intestinalis* the meronts appear to be transient and replication appears to occur primarily during the sporont stage[5,22,23]. In line with this, we are unable to identify any meronts in our dataset. In PVs at 48 hpi, we observe more parasites in the later stages of development. The majority of parasites are spores (69%), with many fewer sporonts (7%) and sporoblasts (24%) (Fig. 2D, G; Supplementary Figs. 1C, D and 2A). In PVs containing only sporonts, these parasites lie very close to the PV membrane, where they are tightly packed and occupy ~80% of the PV volume (Fig. 2E; Supplementary Fig. 2B, C; Supplementary Movie 1). However, in larger PVs containing a mixture of parasites at different stages of development, large spaces are present between the developing parasites, and only

~40% of the PV is occupied by parasites (Fig. 2F, G; Supplementary Fig. 2B, C). When spaces are observed between parasites, they are often filled with vesicle-like structures that may be either secreted from the developing parasites or internalized from the host cytosol (Supplementary Fig. 2D, E). It is possible that similar to other intracellular parasites[41–43], the uptake of vesicles from the host could be one mechanism of acquiring nutrients across the vacuole membrane. Aside from vesicles, we do not observe any discrete structures in interparasite spaces, which is in contrast to previous TEM data that shows the presence of a dense matrix between parasites[5,22]. This difference could be for several reasons, including differences in sources of our samples, or techniques used for imaging.

### 3D snapshots of *E. intestinalis* parasite development

To better understand how organelles are built and organized within the developing spore, we first classified parasites observed in our SBF-SEM datasets as sporonts, sporoblasts, or spores based on previous 2D TEM analyses of infected host cells[5,22,23]. We then segmented the parasite surface (plasma membrane or spore coat), in order to define the boundaries of each parasite, as well as any discernible internal organelles. From our segmentation analysis, we generated 3D reconstructions for parasites from each developmental stage.

**Sporonts.** All 17 of our reconstructed PVs contained at least 1 sporont, the earliest developmental stage observed in our datasets. We analyzed a total of 64 sporonts across the 17 PVs, from which we generated 3D reconstructions of 12. Our 3D reconstructions revealed that sporonts are irregular, flattened structures. The flattened appearance of sporonts may arise from the absence of most organelles at this stage, as well as from the flat shape of the PV. We observed sporonts as either single mononucleated cells (9 out of 64 cells; 9 nuclei), or more often as chains of semi-cellularized, multinucleated cells that occurred in multiples of 2 (55 out of 64 cells, containing a total of 162 nuclei) (Fig. 3A). Multinucleated sporont chains with two nuclei were observed most frequently (33 out of 64 cells; 66 nuclei), in addition to chains of four nuclei (20 out of 64 cells; 80 nuclei) and eight nuclei (2 out of 64 cells; 16 nuclei). The observation that the number of nuclei in sporont chains follows $2^n$ for $n = 0–3$ suggests that genome replication and cytokinesis are synchronized within each chain. The parasite nucleus and a dense mesh of ER-like membrane were the only organelles discernable in our datasets at this stage of development (Fig. 3B; Supplementary Fig. 3A; Supplementary Movie 2). On average, sporonts are 2.3 $\mu m^3$ in volume and ~3.5 $\mu m$ in length along the anterior-posterior (A-P) axis (Fig. 3C; Supplementary Fig. 4A).

**Sporoblasts.** In addition to sporonts, six of the 17 PVs we reconstructed also contain sporoblasts, the next stage of development. In the sporoblast stage, cellularization has completed and the development of specialized parasite organelles begins. We analyzed 61 sporoblasts across all the PVs and generated 18 3D reconstructions. Following cell division, each sporoblast contains a single nucleus and is enveloped in its own plasma membrane (Fig. 3B; Supplementary Fig. 3B–D). At this stage of parasite development, the spore coat is not fully developed: the exosore layer is present, but the endospore layer is not discernable (Supplementary Fig. 3B–D). In addition to the parasite nucleus and a dense mesh of ER-like membrane, this is the first stage of development in which the PT and posterior vacuole are observed. Since the PT length increases as the parasite develops, up to a final length of ~41 $\mu m$ in mature spores, we used the increasing length of the PT to classify sporoblasts into three stages. Stage 1: PT length less than 10 $\mu m$; Stage 2: PT length 10–15 $\mu m$; and Stage 3: PT length greater than 15 $\mu m$ (Fig. 3B, D; Supplementary Movies 3–5). The average length of a sporoblast is 3 $\mu m$ along the A-P axis across all three stages (Supplementary Fig. 4A). However the sporoblast volume changes from 1.8 $\mu m^3$ in Stage 1 to 1.3 $\mu m^3$ in Stage 3 (Fig. 3C). Across

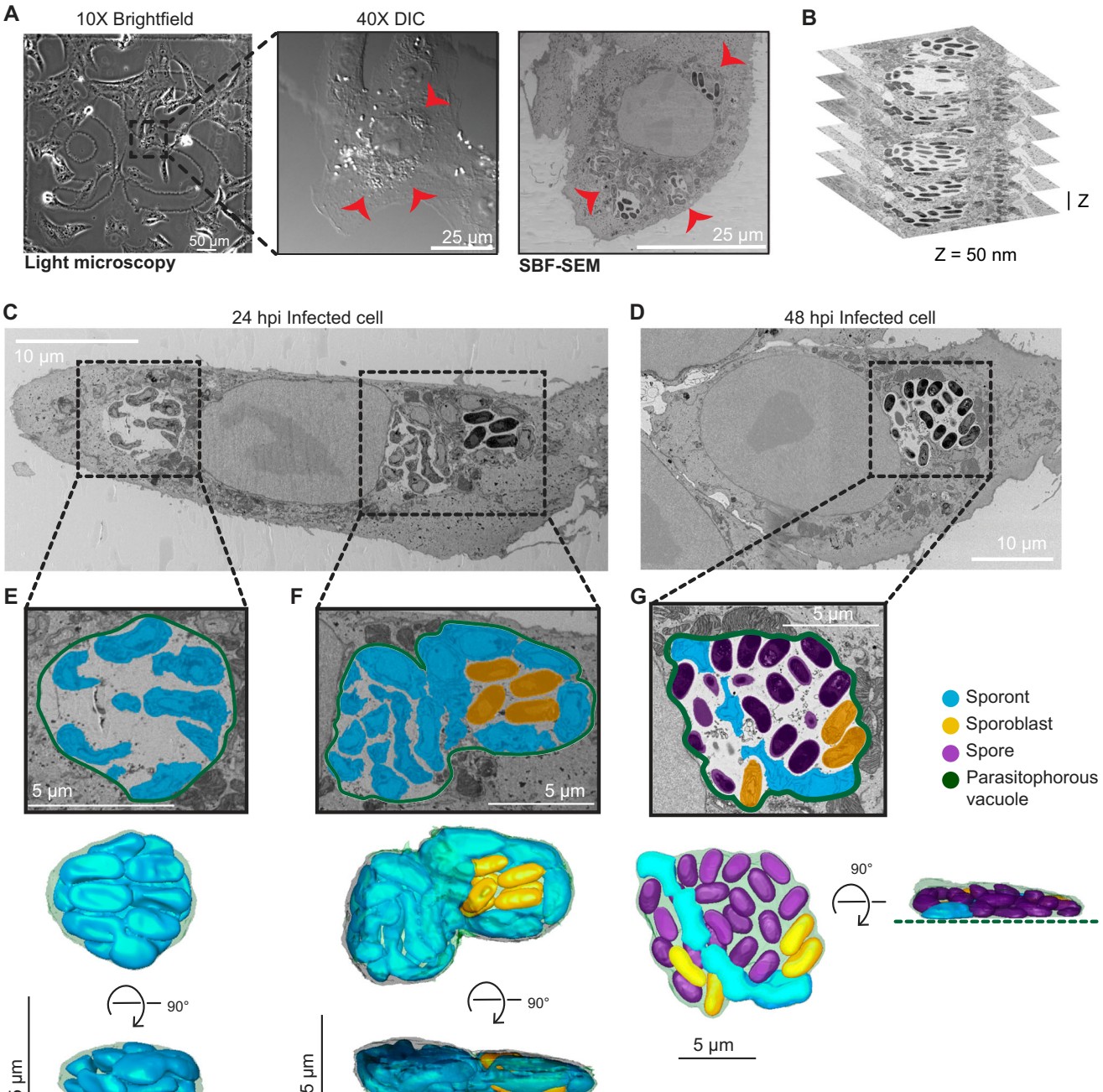

**Fig. 2 | SBF-SEM of Vero cells infected with *E. intestinalis*. A** Representative brightfield, differential interference contrast (DIC) and SBF-SEM micrographs of an infected cell at 24 hpi. The black dashed box indicates the infected cell in the corresponding higher magnification DIC micrograph (middle) and slice of SBF-SEM data (right). Red arrowheads indicate the parasitophorous vacuoles identified in a light microscopy image (middle) and in a slice of the SBF-SEM data (right). Datasets for two infected cells were acquired at 24 hpi, and two infected cells were acquired at 48 hpi. **B** Representative slices of SBF-SEM data, which were acquired serially with a Z-depth of 50 nm. **C, D** Representative slices from SBF-SEM datasets of infected cells at 24 hpi (**C**) and 48 hpi (**D**). Black dotted boxes indicate parasitophorous vacuoles that contain replicating *E. intestinalis* parasites at different stages of development. **E–G** Segmented parasitophorous vacuoles 24 hpi (**E, F**) and 48 hpi (**G**). Data for the vacuole in (**G**) is incomplete in our dataset, which did not capture the whole cell, as indicated by the dotted green line.

these stages, the developing PT is associated with an electron-dense organelle at one end, which we term the nucleation center due to a possible role in nucleating the growth of the PT (Fig. 3B; Supplementary Fig. 3B–D). Interestingly, previous work on *Nosemoides vivieri* also noted the presence of a morphologically similar structure, termed the sac polaire[44], which is sometimes associated with the PT and Golgi-like vesicles. Strikingly, the overall shape of parasites transforms from flatter, elongated structures in Stage 1 to more 3-dimensional, bean-shaped structures by Stage 2, which may be related to the assembly of the PT.

**Spores.** Five of the six PVs from 48 hpi contain spores, the final intracellular stage of parasite development. We observed 140 spores across all PVs analyzed and generated 3D reconstructions of 12 spores. The spores are bean-shaped, ~2.5 μm in length along the A-P axis, and 1.26 μm³ in volume (Fig. 3C; Supplementary Fig. 4A). In addition to the organelles present in the earlier stages of parasite development, in the maturing spore, we also observe (1) clear presence of the endo-spore layer between the exospore and the plasma membrane, and (2) the lamellar polaroplast at the anterior end of the spore, where it surrounds the linear segment of the PT (Fig. 3B; Supplementary Fig. 3E;

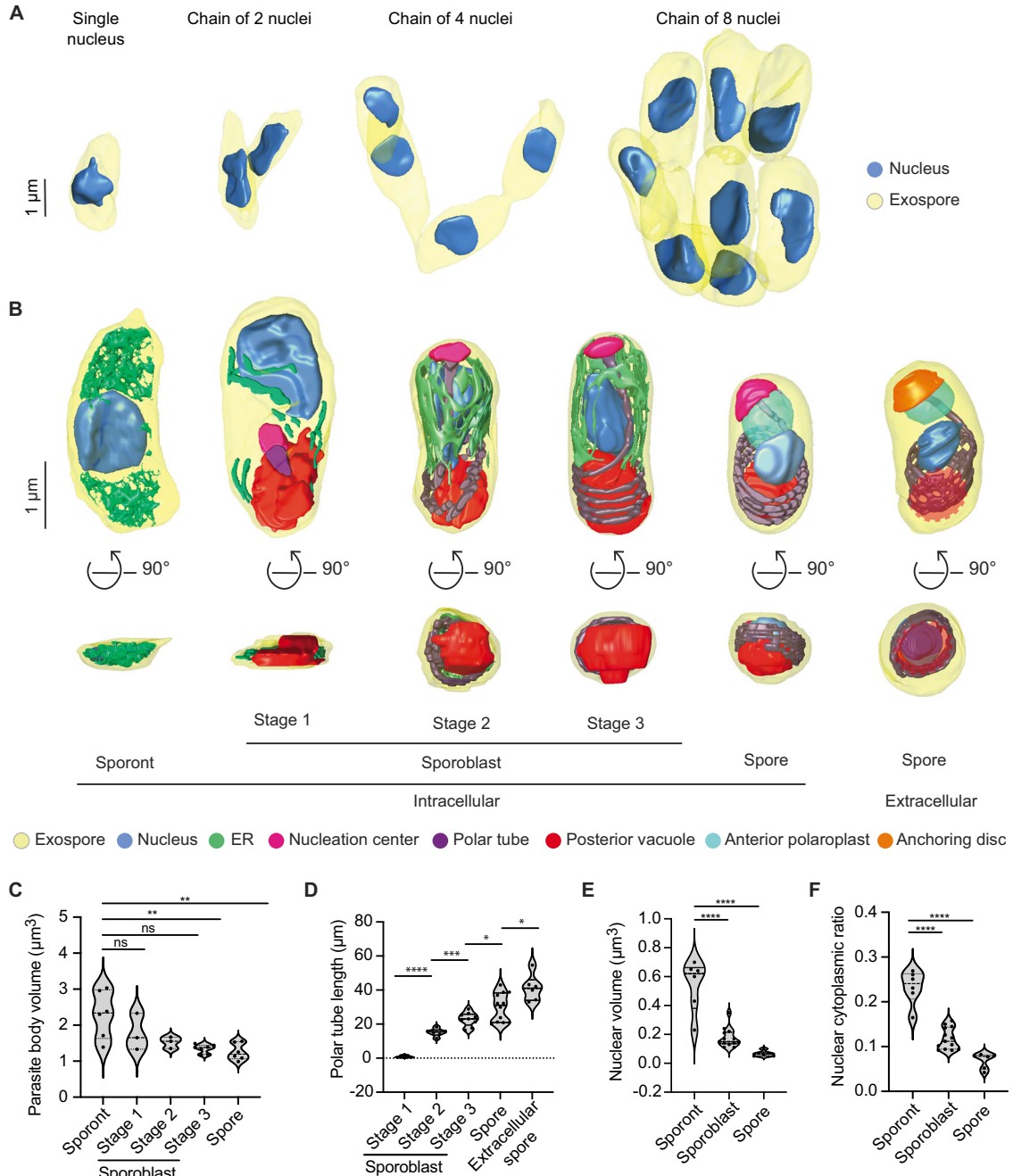

**Fig. 3 | 3D reconstructions and analysis of *E. intestinalis* parasites at different stages of development. A** 3D reconstructions of sporont chains containing 1, 2, 4 and 8 nuclei. **B** Representative 3D reconstructions of each *E. intestinalis* developmental stage. Posterior vacuole in extracellular spore indicated by dotted red circle. **C** Quantification of parasite volume at each developmental stage. **D** Quantification of polar tube length at each developmental stage. **E** Quantification of nuclear volume at each developmental stage. **F** Nuclear-cytoplasmic ratio calculated for each developmental stage. Violin plots show median and quartiles, $*p < 0.05$, $**p < 0.01$, $***p < 0.001$, $****p < 0.0001$; ns, not significant ($p > 0.05$), calculated from an unpaired Student's *t*-test for (**C**–**F**). Source data for (**C**–**F**) are provided in the Source Data file.

Supplementary Movie 6). However, compared to previous 3D spore reconstructions of the closely related *Encephalitozoon hellem*[16], the anchoring disc is not visible in *E. intestinalis* spores observed in the PV at 48 hpi, and the endospore layer appears to be thinner. These observations, combined with the fact that the PVs still contain a significant percentage of parasites at earlier developmental stages, lead us to speculate that the spores within these PVs may not be fully mature. For comparison to fully mature, extracellular spores, we purified *E. intestinalis* spores after several days of infection in Vero cells and collected an SBF-SEM dataset. Overall, the appearance of purified

spores is very similar to the spores observed within the PV. However, consistent with our hypothesis, the endospore layer is thicker in purified spores (~52 nm, vs. ~24 nm for spores within PVs), and the PT is clearly capped by the anchoring disc at the spore anterior (Fig. 3B; Supplementary Fig. 3F; Supplementary Movie 7). These observations suggest that the anchoring disk is one of the last organelles to develop in the spore. In addition, in purified spores, we do not observe a density corresponding to the texture of the density we have annotated as the nucleation center. This could be because (1) the nucleation center no longer remains once the PT is fully developed; (2) the

nucleation center is not visible due to technical issues such as imperfect stain penetration; or (3) the nucleation center matures into the anchoring disc. As the shape of the anchoring disc and the nucleation center are strikingly similar, it is likely that the nucleation center matures into the anchoring disc, but that its composition changes, leading to a change in the appearance of the density.

In addition to the changes in parasite shape observed during development, we also found that parasite volume decreases from 2.29 $\mu m^3$ in the sporonts to 1.3 $\mu m^3$ in spores. Simultaneously, we observed a substantial decrease in nuclear volume, from 0.54 $\mu m^3$ in sporonts to 0.06 $\mu m^3$ in spores (Fig. 3E). The nuclear volume decreases more rapidly than that of the cytoplasm, such that the nuclear-cytoplasmic ratio (NC ratio) changes throughout development, from 0.19 in sporonts to 0.05 in spores (Fig. 3F). The NC ratio is usually constant in a given cell type and species[45,46], and maintaining a constant NC ratio is tightly regulated both by biological factors[47–51] and biophysical forces like osmotic pressure[52]. However, the NC ratio can

vary considerably between cell types, and consequently, is known to change during cellular differentiation[46]. The reduction in NC ratio during *E. intestinalis* development could be the result of the spore becoming more metabolically dormant as it matures. Moreover, minimizing the NC ratio may allow for the nucleus to be small enough to be packaged and squeezed into the much narrower polar tube[16], to be transported into a new host cell during infection.

## Polar tube development and organization

Our 3D reconstructions provide new insights into how the PT organelle develops. Globally, the PT consists of a linear segment at the apical end, and multiple coils toward the posterior end of a mature spore (Fig. 4A). The question of how the PT develops has remained unanswered. Given that the length of the PT increases as the parasite matures, possible models of PT development include: (1) the coiled portion of the PT develops first, followed by assembly of the linear segment, (2) the linear segment of the PT develops first, followed by

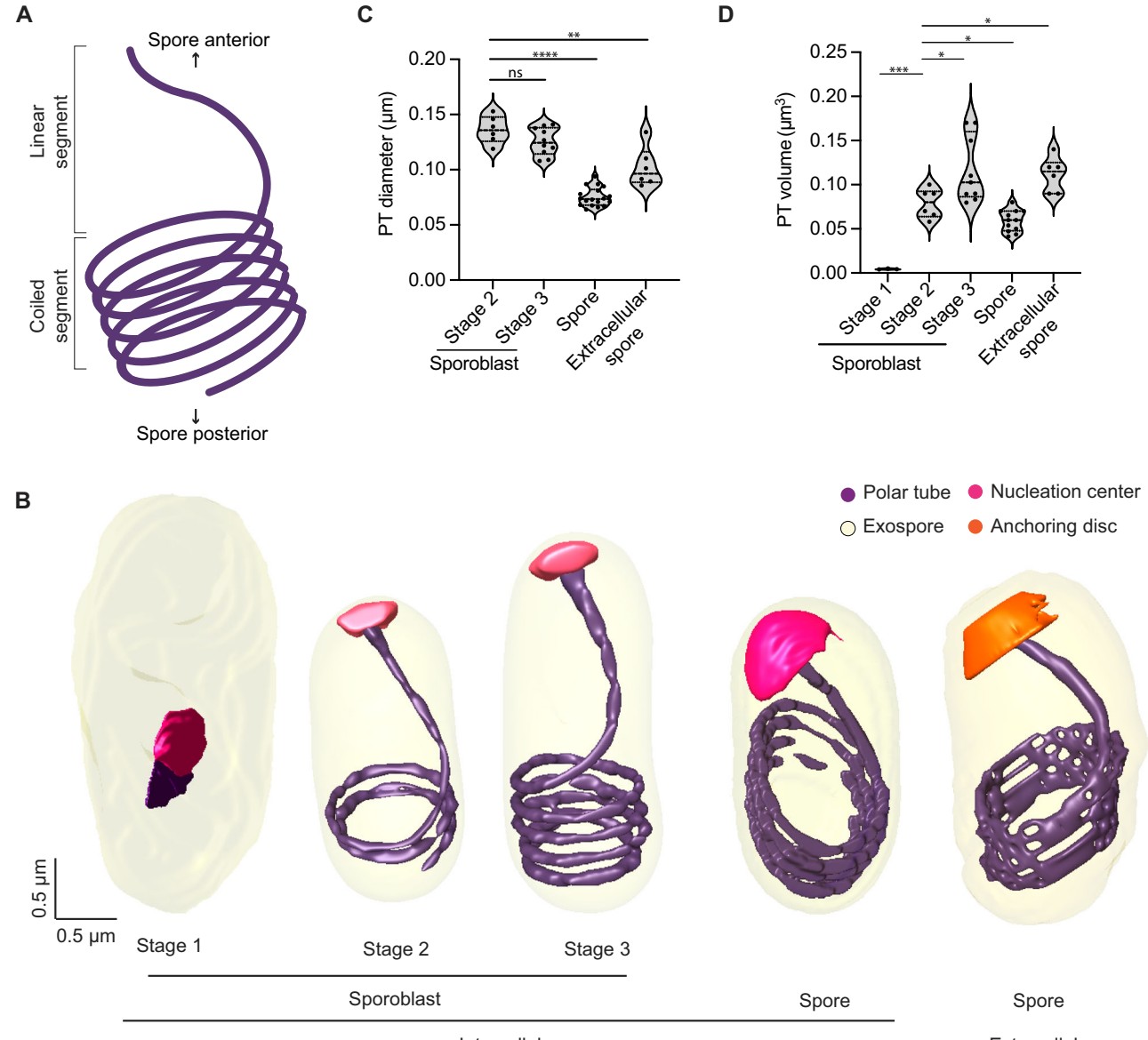

**Fig. 4 | Polar tube development across different developmental stages of *E. intestinalis*. A** Schematic illustrating the global organization of the polar tube, which contains a linear segment toward the anterior end of the spore and a coiled segment toward the posterior end. **B** Representative 3D reconstructions of the polar tube at each *E. intestinalis* developmental stage. **C** Quantification of polar tube diameter at each developmental stage. **D** Quantification of polar tube volume at each developmental stage. Violin plots show median and quartiles, *$p < 0.05$, **$p < 0.01$, ***$p < 0.001$, ****$p < 0.0001$; ns, not significant ($p > 0.05$), calculated from an unpaired Student's *t*-test for (**C**, **D**). Source data for (**C** and **D**) are provided in the Source Data file.

assembly of the coiled region or (3) preformed fragments of the PT coalesce at multiple junctures to form the PT[44]. With any of these models, it is unclear where the oldest and newest portions of the tube are. Here, our 3D models of the PT in different stages of development, provide some insights into the PT assembly process.

The first stage in which we can clearly identify the PT is Stage 1 sporoblasts, in which the PT is visible as a very short linear segment attached to the nucleation center (Fig. 4B). The nucleation center remains associated with the PT throughout its development, and it is possible that this electron-dense organelle serves as a site where new PT material is assembled. The sac polaire, observed previously in *N. vivieri* may be related to the nucleation center we observe. Although the texture of the density is different, the shape of the organelle is often similar. Previously, the sac polaire was often observed adjacent to Golgi-like vesicles, which were hypothesized to be the source of the PT[44]. In our SBF-SEM data, we are unable to identify the Golgi unambiguously. Focusing on the PT, we observed very few sporoblasts with short linear segments (Stage 1 sporoblasts; 4 out of 61), suggesting that this stage of PT growth is transient, and development rapidly proceeds to the next observed stage, in which the linear segment of the PT is substantially longer. As the PT length increases, the number of coils observed at the posterior end of the cell also increases. Stage 2 sporoblasts have 1–2 coils at the posterior end (24 out of 61), while more mature, Stage 3 sporoblasts have 3–4 coils (33 out of 61) (Fig. 4B). The appearance of PT coils in Stage 2 sporoblasts coincides with changes in the shape of the parasite. As the PT is fairly stiff on the micron-length scale, the coiling of the growing PT could serve as a mechanical scaffold, or ribcage, giving structure to the spore body. The PT coils may reduce the deformability of the parasites in later developmental stages, and provide their characteristic shape. We did not observe parasites in which the coiled portion of the PT exists without a linear segment, supporting a model of development in which the linear segment is formed first, followed by the progressive addition of coils as the parasites mature.

Similar to the PT organization in *E. hellem* spores and *Anncaliia algerae*[16], the PT in *E. intestinalis* always forms a right-handed helix (Supplementary Fig. 4B). The *E. intestinalis* PT is anchored to the anterior end of the spore wall, slightly off-center with respect to the A-P axis (Supplementary Fig. 4C), as previously observed in *E. hellem*[16]. Although PT handedness is established early in parasite development (Stage 2 sporoblasts), the off-center anchoring of the PT with respect to the A-P axis is only observed in spores (Supplementary Fig. 4C). This change in PT positioning coincides with the development of (1) the endospore layer and (2) the lamellar polaroplast that snugly surrounds the linear segment of the PT. We also calculated the diameter and volume of the PT at different stages of parasite development (Fig. 4C, D). While PT length increases during development, its diameter decreases, with a 1.7-fold reduction in diameter from Stage 3 sporoblasts to spores (Fig. 4C). Given that the PT is composed of multiple layers[53,54] (Supplementary Fig. 4D), it is unclear from the resolution of our SBF-SEM data whether these changes in PT diameter arise from changes in the proteins hypothesized to form the structural skeleton of the PT[55–58], or from changes in the surrounding elements, such as membranes.

## *E. intestinalis* niche in host cells

The PV membrane forms a boundary between the replicating parasites and the host cytoplasm. This boundary likely helps *E. intestinalis* avoid detection by host innate immune sensors while facilitating the acquisition of metabolites from the host. Our 3D reconstructions of infected cells allowed us to explore the physical interactions between PVs and host organelles. We segmented all discernible host cell organelles from the four infected cells described above, including the nucleus, mitochondria and ER (Fig. 5A, B; Supplementary Fig. 5A–F). To quantify interactions between the PV and host organelles, we scored two

segmented organelles as forming a 'potential contact site' if the distance between them was equal to or less than 30 nm. We refer to these interactions as potential contact sites since the resolution of our SBF-SEM datasets does not allow us to ascertain if they physically interact (e.g., via protein-protein or protein-lipid interactions), or are merely in very close proximity. The largest number of potential contact sites (42% of all sites) are observed between the PV and the host mitochondria (Supplementary Fig. 5M). Potential contact sites can also be found between the ER and PV (33% of all sites). However, upon inspection of 3D-reconstructed cells at both 24 hpi and 48 hpi, it appears that the ER is not selectively recruited to the PVs, but rather is surrounding all organelles (Fig. 5A, B; Supplementary Fig. 5E, F). Potential contact sites are also observed between the nucleus and some PVs (25% of all sites). In infected cells with multiple PVs, at least one PV per cell abuts the host nucleus, but many do not (Fig. 5C, D; Supplementary Fig. 5G, H). When PVs are in close contact with the nucleus, we observe small indentations in the nuclear membrane (Supplementary Fig. 5L), consistent with previous 2D TEM images of Caco-2 cells infected with *E. intestinalis*[31]. Since PV proximity to the host nucleus is highly variable, we asked whether PVs containing specific parasite stages are more likely to form potential contact sites with the host nucleus. However, upon examining the composition of each PV, we did not find any correlation between the parasite developmental stage and the proximity of the PV to the nucleus (Fig. 5G, H; Supplementary Fig. 5K, M). The variability in the position of PVs relative to the host nucleus, coupled with the lack of any trend related to the parasite developmental stage, suggests that the proximity of the PVs to the host nucleus may simply reflect spatial constraints within the cell, rather than a specific interaction. In contrast to the positioning of PVs relative to the host nucleus, we observed that all PVs form potential contact sites with host mitochondria regardless of the parasite composition of the PV (Fig. 5E, F; Supplementary Fig. 5I, J, M).

The consistency of the observed PV-mitochondria potential contact sites suggests that the interaction with host mitochondria is likely important for all parasite developmental stages, and unlike the ER and nucleus, host mitochondria appear to be selectively recruited to PVs, consistent with previous observations[29–31]. Given that microsporidia have highly reduced genomes, it is possible that these physical contacts with mitochondria allow the parasite to rapidly co-opt host metabolites for successful replication and development. In addition to physical proximity, our 3D reconstructions hinted at morphological changes in mitochondria in cells infected with *E. intestinalis*. To better understand these morphological changes, we segmented and compared the shape of individual mitochondria between a 24 hpi cell and an uninfected control. For each individual mitochondrion (see "Methods"), we then calculated an aspect ratio, which allowed us to assess if a mitochondrion was more elongated or more spherical. In uninfected cells the mitochondrial network is composed of consistently longer, linear, filamentous mitochondria (Fig. 6A, B; Supplementary Fig. 6A). In contrast, in infected cells, the mitochondrial network is composed of shorter and more spherical mitochondria (Fig. 6A, B; Supplementary Fig. 6A). Together, our data show that all PVs are in close contact with mitochondria, and the morphology of the mitochondrial network is altered, resulting in spheroid-shaped mitochondria[59].

## The dynamics of host cell mitochondrial remodeling during *E. intestinalis* infection

Our SBF-SEM analysis of host cells infected with *E. intestinalis* is limited in throughput to only a few cells. To assess whether mitochondrial localization and fragmentation are commonly observed in infected Vero cells, we turned to optical microscopy for increased throughput. We analyzed *E. intestinalis* infection in Vero cells at 6, 12, 18, 24, 30, 36 and 48 hpi. To identify early infection events in host cells, we used an RNA fluorescence in situ hybridization (FISH) probe for *E. intestinalis*

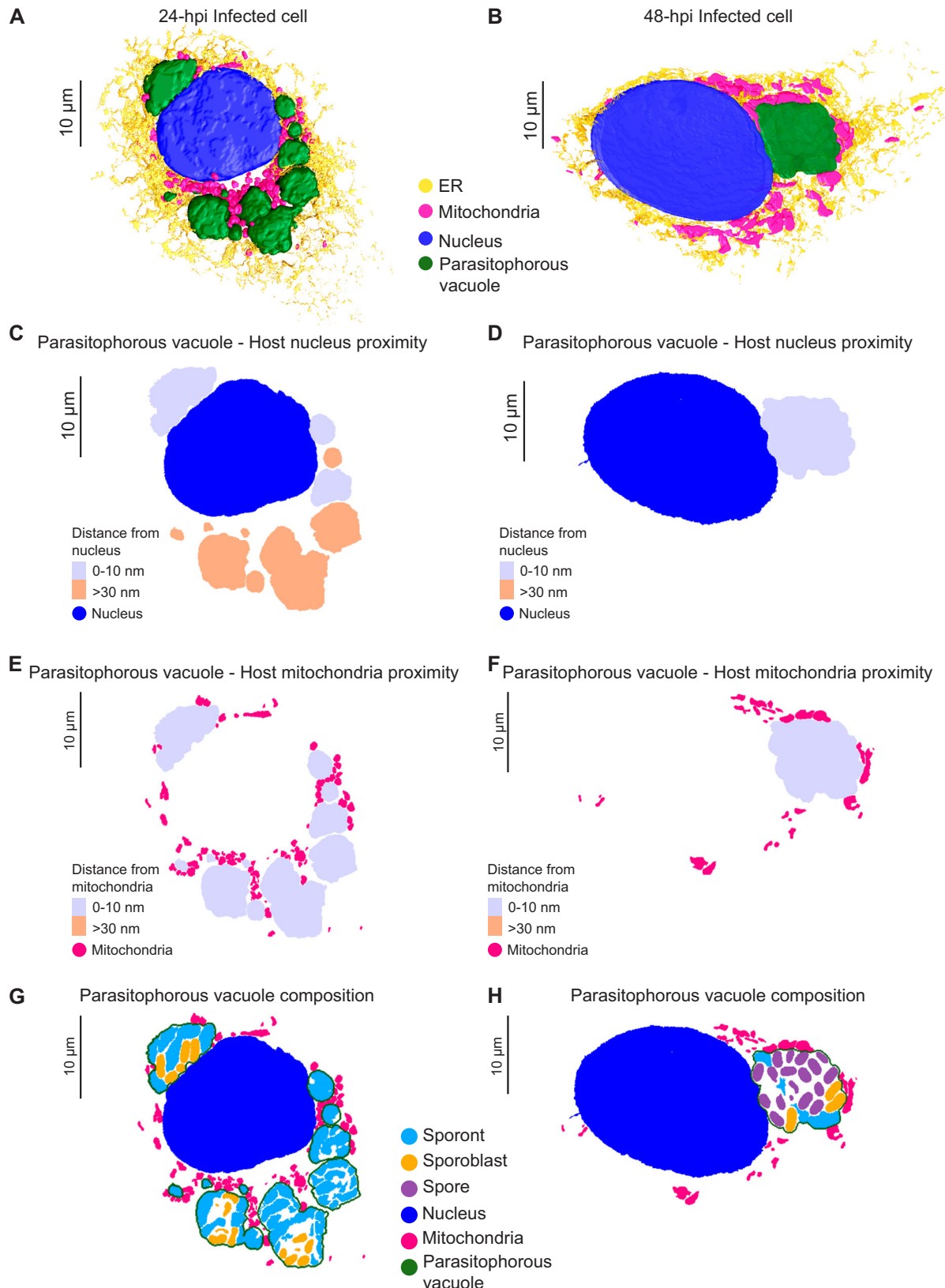

**Fig. 5 | Host cell niche in Vero cells infected with *E. intestinalis*. A, B** 3D reconstructions of Vero cells infected with *E. intestinalis* at 24 hpi (**A**) or 48 hpi (**B**). **C, D** Distance maps showing the minimal distances of parasitophorous vacuoles from the host nucleus at 24 hpi (**C**) and 48 hpi (**D**). **E, F** Distance maps showing the minimal distances of parasitophorous vacuoles from the host mitochondria at 24 hpi (**E**) and 48 hpi (**F**). **G, H** Slices through 3D reconstructions of the intracellular niche showing parasitophorous vacuole composition at 24 hpi (**G**) and 48 hpi (**H**).

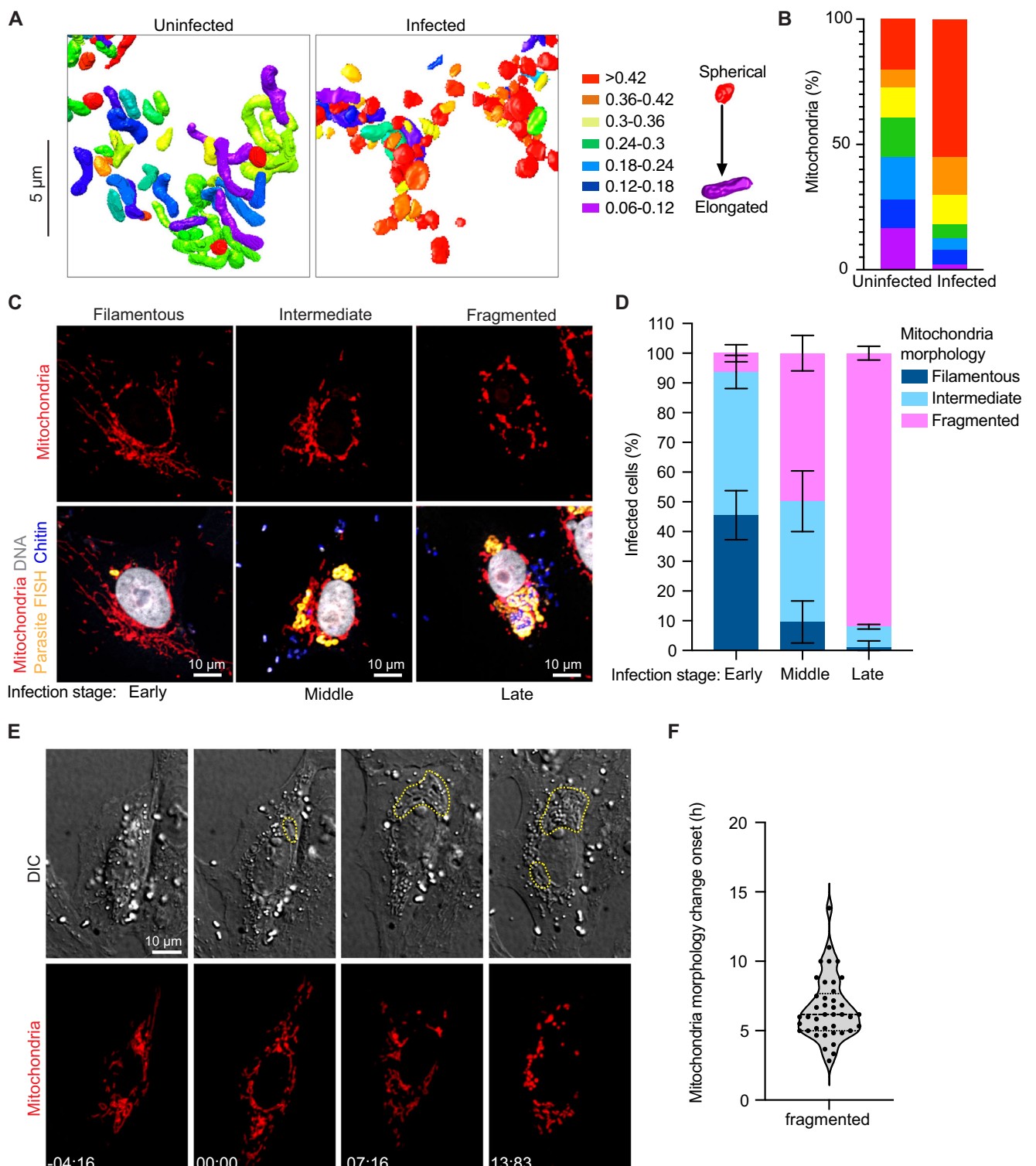

**Fig. 6 | Host mitochondria remodeling during *E. intestinalis* infection. A** 3D reconstructions of host mitochondria in an uninfected and infected cell, colored by aspect ratio, and **B** Quantification of mitochondrial aspect ratio from a given volume of an uninfected and infected cell. $n \geq 100$ mitochondria from $n = 2$ uninfected and infected cells. **C** Vero cells infected with *E. intestinalis*, fixed and analyzed by *E. intestinalis* RNA FISH (yellow) and stained for DNA (DRAQ5, gray), chitin (calcofluor white, blue) and mitochondria (Tom70 antibody, red). Representative micrographs of infected cells and their host mitochondria morphology (top panel) and the corresponding *E. intestinalis* infection stage (bottom panel) are shown. **D** Quantification of mitochondrial morphology (filamentous, intermediate, or fragmented) in early, middle and late *E. intestinalis* infected Vero cells. Data are represented as mean ± SD from three biological replicates. $n \geq 100$ cells per experiment. **E** Vero cells expressing DsRed targeted to the mitochondrial matrix were infected with *E. intestinalis* and imaged for 40 h. Differential interference contrast (DIC) (top montage) was used to monitor infection and Mito-DsRed (bottom montage) was used to monitor host mitochondria remodeling. Infection clusters in cells are marked with a yellow dotted line (top montage), and white arrowheads indicate fragmented mitochondria around the infection clusters (bottom montage). Representative montage from 4 independent experiments. **F** Mitochondria fragmentation onset is plotted; all experiments yielded similar results. Violin plot shows median and quartiles, $n = 41$ cells from 4 independent experiments. Source data for (**B**, **D** and **F**) are provided in the Source Data file.

16S rRNA, as well as a DNA stain to monitor both host and parasite DNA, and an antibody against Tom70 to monitor host mitochondria. Our data show that 100% of PVs (559 PVs across 311 infected Vero cells) are co-localized with mitochondria, consistent with the observation from SBF-SEM, and previous TEM analysis[29–31]. In some cases, PVs are entirely enmeshed within the mitochondrial network (72.6% of infected cells), while in other cases, PVs contact fewer mitochondria (27.4% of infected cells) (Supplementary Fig. 6B).

To analyze host mitochondrial fragmentation in infected cells, we classified the mitochondria into three categories: (1) filamentous, in which the mitochondria are elongated and tubular; (2) intermediate, in which the mitochondrial network is composed of a mixture of tubular and spherical structures; and (3) fragmented, in which the mitochondria are mainly composed of individual, punctate structures (Fig. 6C). Our analyses showed that at 36 hpi, ~48% of infected cells had fragmented mitochondria, compared with 16% of uninfected cells (Supplementary Fig. 6C), consistent with a recent observation in other *Encephalitozoon spp*[32]. We next asked how fragmentation correlates with the stage of infection. We used the RNA FISH probe together with calcofluor white, a chitin dye, to stage infected cells into three categories: (1) early-stage infections, vacuoles containing single FISH puncta only; (2) middle-stage infections, vacuoles containing chains of FISH puncta or (3) late stage infection, vacuoles containing chitin positive parasites, indicating development of a spore coat. In early-stage infections, we observe ~6% of cells with fragmented mitochondria, while at late-stage infections, we observe ~90% of cells with fragmented mitochondria (Fig. 6D). A closer examination of the fragmented mitochondrial networks in the infected cells revealed that in ~55% of late-stage infected cells, the mitochondria were spheroids[59], consistent with our SBF-SEM data (Supplementary Fig. 6D). Such morphological changes to mitochondrial shape have been linked to increased reactive oxygen species (ROS) and a loss of mitochondrial membrane potential[60,61], which can trigger apoptosis. To examine if the observed changes in mitochondrial morphology were a precursor to the induction of apoptosis, we quantified the percentage of cells with chromatin condensation, by observing the size and intensity of nuclear staining. Apoptotic chromatin condensation was observed in only ~5% of late-stage infected cells and was not observed in cells with early or middle-stage infections, or in uninfected cells (Supplementary Fig. 6E). As apoptotic nuclei were rarely observed, it is likely that despite the changes in mitochondrial morphology, most late-stage infected cells are not apoptotic.

To better understand the dynamics of fragmentation, we used live-cell imaging. We infected Vero cells expressing a mitochondrial matrix-targeted DsRed[62] with *E. intestinalis* spores and monitored changes to the host mitochondrial network during infection by time-lapse confocal microscopy. Probes for live-cell imaging of *E. intestinalis* are not available, making it challenging to identify single infection events or very early stages of infection. However, we can readily identify parasites within infected cells by differential interference contrast (DIC) microscopy in the later frames of our live-cell imaging experiments. Once these later infection stages are identified by DIC, we can trace these infection events back to earlier frames of our movies, and identify a window of time in which *E. intestinalis* infection likely occurred. At these earlier infection time points (~6 hpi), the mitochondrial network is composed of filamentous mitochondria (Fig. 6E; Supplementary Movie 8). As the parasites begin to replicate, we observe localized changes in the morphology of mitochondria in close proximity to the PV, from filamentous to shorter and less tubular. As the parasites undergo multiple rounds of replication, these morphological changes in the mitochondria spread through the entire host mitochondrial network. The change in mitochondrial network morphology occurs over a 6.5 h ± 0.35 time frame (Fig. 6E, F; Supplementary Movie 8). Overall, this suggests that the change in the mitochondrial network in infected cells is (1) a response to infection

and (2) triggered locally by a signal emanating from the parasites, and spreads over time to more distal mitochondria. These initial localized changes in the mitochondrial network may result from local depletion of metabolites by the parasites and/or secretion of parasite proteins or other factors locally into the host cytosol. The global fragmentation of the mitochondrial network could result from a more generalized host stress response.

From our live-cell imaging datasets, we were also able to capture 12 examples of infected cells undergoing cytokinesis, allowing us to assess how the parasites are segregated to the daughter cells, and the impact on the mitochondrial network. In 5 of the 12 dividing cells, the PVs containing replicating parasites were segregated only to a single daughter cell. This suggests that the segregation of parasites to the two daughter cells may be random. Moreover, although we cannot rule out the possibility that the parasites may be actively partitioned, if such a mechanism exists, it appears to be imperfect. Intriguingly, in the five-cell division events where the parasites are partitioned into a single daughter cell, the mitochondrial network in the infected daughter is fragmented, while the uninfected daughter regains a filamentous mitochondrial network (Supplementary Movie 9). Combining our dynamic light microscopy data at lower resolution with static snapshots from SBF-SEM data at higher resolution, we conclude that the mitochondrial network is substantially remodeled, leading to fragmentation in response to *E. intestinalis* infection, and that the mitochondrial remodeling is reversible if the infection is lost from the cell.

## Discussion

Combining our data from SBF-SEM and light microscopy of Vero cells infected with *E. intestinalis* parasites, we propose a refined model for parasite development within vacuoles, and parasite organelle development through the life cycle (Fig. 7A; Supplementary Movies 10 and 11). Our model builds on previously published work[5,22,23], incorporating new 3-dimensional information on parasite developmental stages, cell shape and size, and parasite organelles, including intermediates of PT development. This work provides key insights into the asynchronous development of microsporidia in parasitophorous vacuoles. In addition to parasite development, we have gained an understanding of the dynamics of parasite-induced mitochondrial remodeling in the host cell.

### Model for PT development

The PT is the most striking organelle observed in all microsporidian species, with a final length that is substantially longer than the spore itself. Our 3D reconstructions of *E. intestinalis* parasites at different developmental stages allow us to propose the following model of PT development (Supplementary Movie 12). At the earliest stage of parasite development captured in our datasets, the sporont stage, we see no sign of the PT (Fig. 7B). PT assembly begins near the middle of the parasite, from a nucleation center during stage 1 of sporoblast development (Fig. 7C). This observation was similar to previous reports of an electron-dense mass in *E. intestinalis*[5] and sac polaire in *N. vivieri*[44]. At this stage, the parasite is relatively flat and has no apparent polarity. As PT growth continues, new protein subunits are likely added at the junction between the PT and the nucleation center, to build the linear segment of the PT first (Fig. 7D). As the linear segment of the PT continues to grow, it encounters the cell periphery, at which point the PT begins to coil under the strain of continued PT assembly within the confines of the sporoblast. Intuitively, this can be observed when threading stiff tubing (analogous to the PT) into a bottle (analogous to the spore) and inserting new tubing at the top (Supplementary Movie 13). Right-handed coiling may arise from the mechanical properties of the PT itself, much as actin filaments and double-stranded DNA coil with a preferred handedness under strain (Fig. 7E; Supplementary Movie 12). It is an attractive possibility that the A-P axis is established at this transition from stage 1 to stage 2

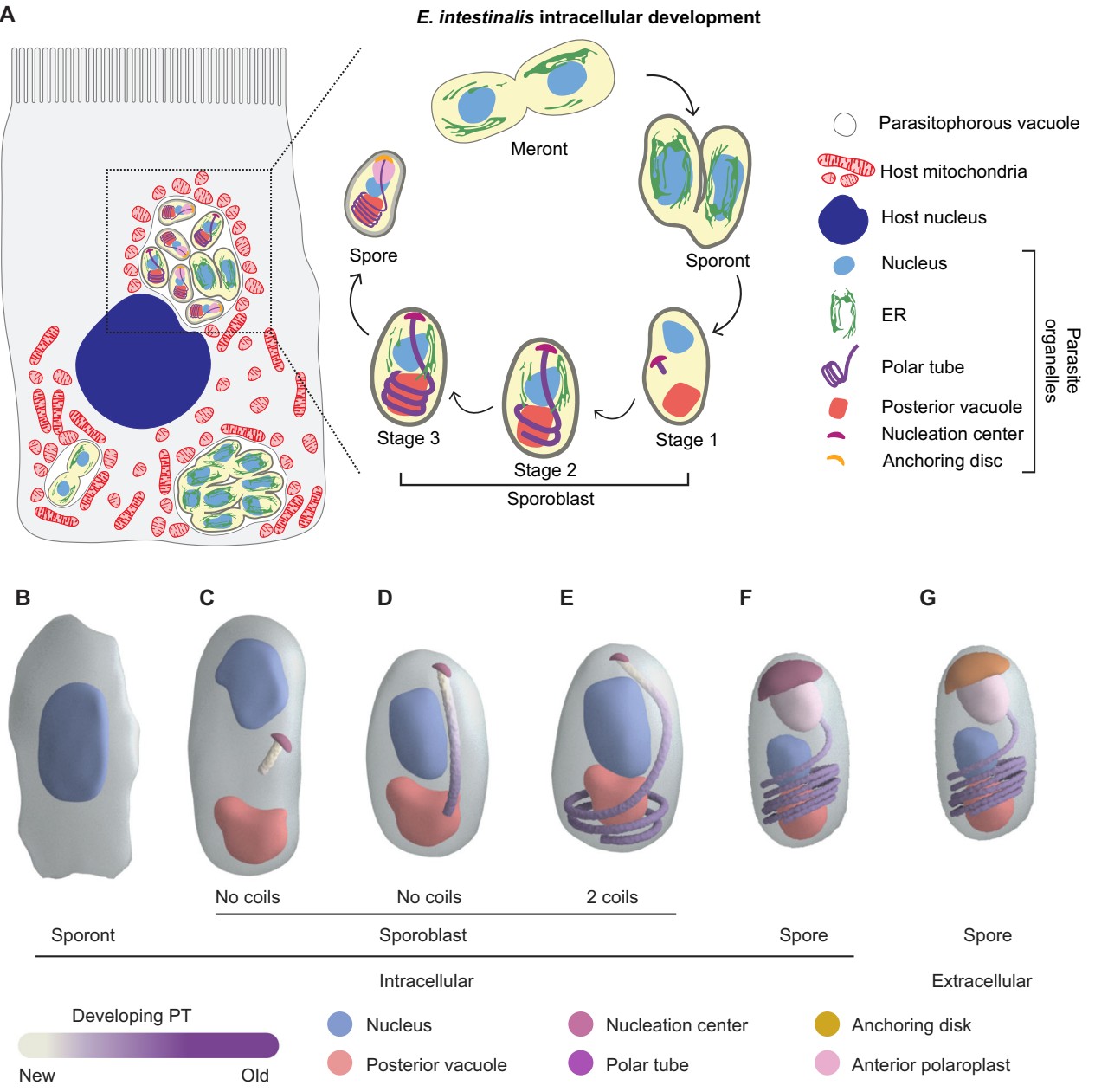

**Fig. 7 | Host niche and intracellular development of *E. intestinalis*. A** Schematic of the host cell niche of *E. intestinalis* and the intracellular parasite developmental stages. **B–G** Model of polar tube (PT) development. **B** The sporont stage, in which the PT has not begun to develop, and the cell is irregularly shaped. **C** The nucleation center is first observed in the Stage 1 sporoblast, toward the center of the cell, from which a small, straight segment of PT starts to grow. The nucleus in these parasites is positioned toward one end, and the posterior vacuole toward the other end. **D** Next, our data suggest that the linear segment of the PT increases in length (this stage is not observed in our dataset, but rather inferred from the preceding and next stages). **E, F** As the PT grows toward the posterior vacuole, it hits the bottom of the spore and begins to coil. The PT continues to grow and coil, during which time its length increases and its diameter decreases, suggesting a combination of polymerization and remodeling are at play during the development process. As the PT grows, we propose that the newest material is at the top, with polymerization occurring at the nucleation center, and the oldest part of the tube toward the bottom of the spore. During this process, (1) the nucleus is repositioned closer to the PT coils, with the PT coils acting as a ribcage around the vacuole and the base of the nucleus, possibly playing a role in giving the cell its shape and (2) the PT coils take on a tilted orientation relative to the A-P axis of the spore, and the anchor point to the spore coat is off-center from the A-P axis. **G** In the mature spore, the anchoring disc is observed, potentially juxtaposed with the nucleation center, functioning to anchor the PT to the spore coat.

sporoblasts, perhaps by the PT itself driving the translocation of the nucleation center to what eventually becomes the anterior end of the spore. Further PT growth during the spore stage results in both an increase in the length of the PT as well as the number of PT coils (Fig. 7F). In dormant, extracellular spores, the PT has 5-6 coils at the posterior end and is capped by the anchoring disc at the anterior end (Fig. 7G).

## Mitochondrial remodeling in infection

The host mitochondrial network is highly dynamic, undergoing cycles of fission and fusion. Fission leads to fragmentation of the mitochondrial network and facilitates the removal of damaged mitochondria by mitophagy, as well as segregation into daughter cells during the cell cycle[63]. Mitochondrial fusion, on the other hand, allows for the exchange of mitochondrial DNA and proteins. Mitochondria play a

central role in metabolism, signaling, and cell death. Consequently, several intracellular pathogens have developed strategies to modulate host mitochondrial morphology and function in order to promote the establishment of an intracellular niche conducive to proliferation. For example, during infection, intracellular pathogens can induce mitochondrial fragmentation to support metabolic rewiring, as in the case of *Legionella pneumophila*[64], cell death in the case of *Helicobacter pylori*[65], or mitophagy in the case of *Listeria monocytogenes*[66]. However, some pathogens induce mitochondrial hyperfusion to promote ATP production for sustaining pathogen proliferation, as in the case of *Chlamydia trachomatis*[67]. Here, we observe substantial host mitochondrial network remodeling in *E. intestinalis*-infected Vero cells, such that the mitochondria are fragmented. Similar fragmentation of host mitochondria has also been observed in cells infected with other *Encephalitozoon spp.*[32]. Our live-cell imaging data revealed that fragmentation of host mitochondria is initially localized to the area surrounding the replicating parasites, and then spreads to the rest of the network. As obligate parasites, *E. intestinalis* has a highly reduced genome, having lost genes for oxidative phosphorylation and needs to siphon ATP from the host. Therefore, it is possible that the observed changes in the host mitochondrial network are a response to the increased energy burden on the host. Alternatively, microsporidia also use a variety of secreted proteins to interact with the host[68]. Therefore, it is possible that this remodeling of the host mitochondrial network is the result of a secreted parasite protein, as is the case for several intracellular pathogens that induce host mitochondrial remodeling[64,69]. Our data provide the first insights into the dynamics of host mitochondrial remodeling in the presence of microsporidia infection. Future studies focused on comparing divergent microsporidian species in different host cell types will reveal exciting new information on the modulation of host mitochondria in response to microsporidian infection, and how mitochondrial remodeling impacts parasite replication and development.

## Methods

### Maintenance of mammalian cell lines

Vero cells (ATCC CCL-81) were grown at 37 °C in a humidified atmosphere with 5% $CO_2$ in DMEM, high glucose growth medium (ThermoFisher Scientific 11965092). Unless specified otherwise, all media was supplemented with 10% fetal bovine serum (VWR Life Science; 89510-188) and non-essential amino acids (Gibco™ 11140050). Cell lines are tested each month using the Invivogen MycoStrip™-Mycoplasma Detection Kit (rep-mys-100) and found negative.

### Propagation of *E. intestinalis* spores

*E. intestinalis* (ATCC 50506) spores were propagated in Vero cells. Vero cells were cultured in Dulbecco's Modified Eagle Medium (DMEM, high glucose) with 10% heat-inactivated fetal bovine serum (FBS) at 37 °C and with 5% $CO_2$. At 70–80% confluence, *E. intestinalis* spores were added into a 75 cm² tissue culture flask, and the media was switched to DMEM, high glucose supplemented with 3% FBS. Cells were allowed to grow for 10 days, with media being changed every 2 days. To purify spores, the infected cells were detached from tissue culture flasks using a cell scraper and placed into a 15 mL conical tube, followed by centrifugation at 1300×*g* for 10 min at 25 °C. Cells were resuspended in sterile DPBS and mechanically disrupted using a G-27 needle. The released spores were purified using a Percoll gradient. Equal volumes (5 mL) of spore suspension and 100% Percoll were added to a 15 mL conical tube, vortexed and then centrifuged at 1800×*g* for 30 min at room temperature. The purified spore pellets were washed three times with sterile DPBS and further purified in a discontinuous Percoll gradient. Spore pellets were resuspended in 2 mL of sterile DPBS and layered onto a 10 mL four-layered percoll gradient (2.5 mL 100% Percoll, 2.5 mL 75% Percoll, 2.5 mL 50% Percoll, 2.5 mL 25% Percoll), and centrifuged at 8600×*g* for 30 min at RT. Spores that separated into the

fourth layer (100% Percoll) were carefully collected and washed twice in 10 mL of sterile DPBS at 3000×*g* for 5 min at RT. Purified spore pellets were stored in sterile DPBS at 4 °C for further analyses.

### SBF-SEM sample preparation

**CLEM samples.** Vero cells were cultured on a gridded glass bottom 35 mm dish (MatTek Corporation, Cat no. P35G-1.5-14-CGRD) in DMEM, high glucose supplemented with 3% FBS for 4 h before adding *E. intestinalis* spores at MOI = 30. We used the primate cell line, Vero, for these experiments, because it supports robust infection and replication of *E. intestinalis*, making it ideal to identify infected cells for SBF-SEM. With this experimental setup, a large percentage of cells are infected, which is critical to enable the identification of infected cells for downstream analysis using SBF-SEM. At 24 hpi and 48 hpi, cell culture media was discarded and 2 mL of fixative (2% PFA and 2.5% glutaraldehyde in 0.1 M sodium cacodylate buffer pH 7.2) was pipetted gently onto the dish. Cells were imaged on a Zeiss AxioObserver microscope, first at high magnification (Plan-Apochromat 40×/1.40 Oil DIC M27 objective) to identify infected cells of interest and record both the cell shape and its precise position relative to the surrounding cells. Next, the cell of interest was imaged at low magnification (EC Plan-Neofluar 10x/0.30 Ph1 WD = 5.2 objective) to create a map of its location showing the gridded lines, and grid numbers or letters that were used as a marker when trimming the EM sample block, and searching for the target cell of interest under EM. The fixative solution was replaced with an equal volume of fresh fixative and incubated at RT for 2 h before continuing the fixation overnight at 4 °C. Samples were then stained as described below.

**Cell pellet samples.** Vero cells were cultured in a 35 mm tissue culture treated dish in DMEM, high glucose supplemented with 3% FBS for 4 h before adding *E. intestinalis* spores at MOI = 20. At 24 hpi and 48 hpi, cell culture media was discarded and 2 mL of fixative (2% PFA and 2.5% glutaraldehyde in 0.1 M sodium cacodylate buffer pH 7.2) was pipetted gently onto the dish. After 1 min of fixation, cells were detached from the dish using a cell scraper. Cells were transferred immediately into a 1.5 mL tube and centrifuged at 2400×*g* for 2 min. The tube was rotated 180° and centrifuged at 2400×*g* for an additional 2 min. The fixative solution was replaced with an equal volume of fresh fixative and incubated at RT for 2 h before continuing the fixation overnight at 4 °C. Samples were then stained as described below.

**Spores purified from Vero cells.** Spore pellets purified from Vero cells were resuspended in 1 mL of fixative (2% PFA and 2.5% glutaraldehyde in 0.1 M sodium cacodylate buffer pH 7.2) and incubated at RT for 2 h. After 2 h, the fixative solution was replaced with an equal volume of fresh fixative and incubated at RT for 2 h before continuing the fixation overnight at 4 °C. Samples were then stained as described below.

**Sample staining post-fixation.** Following fixation, samples were stained and processed for SBF-SEM using a published rOTO protocol[70]. Briefly, samples were stained in sequence in freshly made reduced osmium solution that contains 2% osmium, 1.5% potassium ferrocyanide in 0.1 M sodium cacodylate buffer, for 1.5 h at RT in the dark, 1% TCH (thiocarbohydrazide) for 20 min at RT, 2% aqueous osmium for 40 min at RT, 1% aqueous uranyl acetate overnight at 4 °C, then lead aspartate solution for 30 min at 60 °C. The samples were then dehydrated in a graded ethanol series before embedding in Durcupan.

### SBF-SEM imaging

Each sample block was mounted on an aluminum 3View pin and electrically grounded using silver conductive epoxy (Ted Pella, catalog #16014). The entire surface of the specimen was then sputter coated with a thin layer of gold/palladium and imaged using the Gatan OnPoint BSE detector in a Zeiss Gemini 300 VP FESEM equipped with a Gatan

3View automatic microtome. The system was set to cut 50 nm slices, imaged with gas injection setting at $2.4 \times 10^{-3}$ mBar with Focus Charge Compensation to reduce electron accumulation charging artifacts. Images were recorded after each round of sectioning from the block-face using the SEM beam at 1.2 keV with a beam aperture size of 20 μm and a dwell time of 1.0 μs/pixel. Following image acquisition parameters were used: (1) Cell pellet 48 hpi (dataset 3), 2 nm pixel size and an axial Z resolution of 30 nm, (2) *en face* 24 hpi (dataset 1 and 2) and 48 hpi (dataset 4, 5, 6 and 7), 2 nm pixel size and an axial Z resolution of 50 nm, (3) Spores purified from Vero cells, 2 nm pixel size and an axial Z resolution of 50 nm (dataset 8 and 9) and (4) Uninfected cells (dataset 10 and 11), 2.5 nm pixel size and an axial Z resolution of 60 nm. Data acquisition occurred automatically using Gatan Digital Micrograph (version 3.31) software. For each dataset, the image stack was imported into Fiji[71] for image processing with a Gaussian blur filter (Sigma = 2.0). The resulting processed image stack was exported as a 16-bit tiff image sequence and uploaded into Dragonfly for segmentation analysis.

## Segmentation analysis

Segmentation of organelles of interest, 3D reconstruction, and quantification of the spore size, spore volume, PT volume, PT length and nuclear volume were performed using Dragonfly software, Version 2022.1.0.1259 for [Windows]. Object Research Systems (ORS) Inc, Montreal, Canada, 2020; software available at http://www.theobjects.com/dragonfly. SBF-SEM sections were either manually aligned or automatically aligned using SSD (sum of squared differences) algorithm in the Dragonfly software prior to segmentation. Graphical representation of *E. intestinalis* spores at different developmental stages, the PT and PVs were performed with Dragonfly software. From all parasitophorous vacuoles across datasets, we were able to observe 64 sporonts, 61 sporoblasts and 140 spores. A minimum of 7 parasites were segmented completely per condition. Due to technical limitations in SBF-SEM data collection, we could reliably only compare absolute values for measurements made within the same dataset to account for any cutting artifacts and/ or slice skipping that occurs during image acquisition. We scored an additional 19 sporoblasts that were not included because they were hard to categorize clearly owing to imaging artifacts that preclude us from clearly identifying the PT across all imaged slices in these sporoblasts.

**Distance maps and potential contact site quantification.** Distance maps between the PV membrane and the membrane of the host organelles in infected and uninfected cells were calculated in the Dragonfly software. To identify potential contact sites between the PV and various host organelles, the range of the distance maps created for each PV was increased to 30 nm. The number of intersection sites between the newly created distance map and the host mitochondria, ER or nucleus were annotated in the Dragonfly software and plotted in GraphPad Prism 9.

**Mitochondria aspect ratio quantification.** A single mitochondrion was defined as a double membrane delimited organelle that (1) could be tracked and segmented through consecutive SEM slices and (2) was spatially distinct from surrounding double membrane delimited organelles. Aspect ratio for individually segmented mitochondria from uninfected and infected cell volumes was calculated in the Dragonfly software. For uninfected cells, where partial volumes were acquired by SBF-SEM, only spatially distinct mitochondria for which the entire volume was captured were included in the analyses.

**PT diameter measurements.** PT diameter measurements were made from 2D images. Each coil of the PT is composed of multiple layers. In our SBF-SEM datasets, we were able to clearly observe two layers, an outer layer (dark density) and an inner layer (lighter density). Three separate diameter measurements were made on the inner layer, from one edge to the diametrically opposite edge and averaged. For each parasite stage, the measurement was made from multiple coils at a 2D cross section that represented the middle of the spore.

**Endospore thickness measurements.** The thickness of the endospore was measured in 2D images of spores found in the PV as well as in extracellular spores purified from Vero cells. The endospore layer (lighter density) is sandwiched between the plasma membrane (darker density) and exospore (darker density). Endospore measurements were made on the lighter density, from the outer boundary of the plasma membrane to the inner boundary of the exospore layer. This measurement was made at six spatially distinct regions of the endospore, in a 2D cross section that represented the middle of the spore.

## Immunofluorescence microscopy

For immunofluorescence microscopy of mitochondria following *E. intestinalis* infection for 6, 12, 18, 24, 30, 36 and 48 h, Tom70 (14528-1-AP) antibody was used. Vero cells on coverslips were washed 1x in PBS-T (0.1% Tween-20) and fixed in 4% PFA in PBS-T (0.1% Tween-20) at 37 °C for 15 min. Coverslips were washed in TBS+0.1% Triton X-100, blocked in TBS+0.5% Triton X-100 with 3% bovine serum albumin (BSA) for 30 min, and stained with primary antibody at 1 μg/mL for 3 h at room temperature. Coverslips were washed in TBS+0.1% Triton X-100, 3x for 5 min at RT before species-specific secondary antibodies conjugated to Alexa Fluor 488 (ThermoFisher Scientific; A11008) were applied for 30 min at 5 μg/mL at RT. Coverslips were then washed in TBS+0.1% Triton X-100, 3x for 10 min at RT and 1x with 500 μL hybridization buffer (900 mM NaCl, 20 mM Tris HCl, 0.01% SDS). Then, 50 μL FISH staining solution (125 nM FISH probe in hybridization buffer) was added per coverslip and incubated for 18 h at 37 °C. An *E. intestinalis* 16S rRNA-specific FISH probe conjugated to Quasar 570 (LGC Biosearch Technologies) was used. Following incubation, coverslips were washed 2x with 500 μL of wash buffer (50 mL hybridization buffer + 5 mM EDTA) for 30 min at 37 °C to remove excess FISH probe. Then, 500 μL calcfluor white (18909-100ML-F)/ DRAQ5 (NBP2-81125-50ul) staining solution (2 μg/μL calcfluor white and 0.5X DRAQ5 in PBS-T (0.1% Tween-20)) was added, and samples were incubated for 30 min at RT. Coverslips were washed 1x with TBS+0.1% Triton X-100 and mounted onto slides with Prolong Diamond antifade (ThermoFisher Scientific; P36965) and sealed. For PV localization experiments, Vero cells were incubated with MitoTracker Deep Red FM (ThermoFisher Scientific; M22426) at 200 nM for 45 min prior to fixation and RNA FISH staining. Nucblue (ThermoFisher Scientific; R37605) was used as per manufacturer guidelines, 2 drops per mL of solution. Samples were imaged on a Nikon W1 spinning disc confocal microscope with a Nikon Apo 60×1.40 Oil objective and dual Andor 888 Live EMCCD cameras. Z-stacks were acquired with 0.3 μm spacing. Maximum intensity projection images were created in the Nikon Elements software. Pseudo coloring and cropping were performed in Fiji[71] and images were assembled in Illustrator (Adobe, San Jose, CA). At least 100 cells were analyzed per condition per experiment.

## Live-cell imaging

Vero cells expressing a mitochondrial matrix targeted DsRed were cultured in a 35 mm ibidi cell culture imaging dish in DMEM, high glucose supplemented with 3% FBS for 4-h before adding *E. intestinalis* spores at an MOI = 30. Six hours post-infection, media in the dish was replaced with MEM, no glutamine, no phenol red (Gibco™ 51200038) supplemented with 3% FBS, sodium pyruvate, glutamine and non-essential amino acids and cells were imaged. Live imaging was performed on a Nikon W1 spinning disc confocal microscope with a Nikon Apo 60× 1.40 Oil objective, equipped with dual Andor 888 Live EMCCD cameras, a Piezo Z stage controller and Nikon Perfect Focus. Cells were maintained at 37 °C and 5% CO$_2$ with a Stage Top Incubator with Flow

Control and a Sub Stage Environmental Enclosure (Tokai Hit). Cells were imaged by DIC and fluorescence microscopy (561 laser). Images were acquired at 60X (Apo 60× 1.40 Oil λS DIC N2 0.13 WD) and recorded on an Andor 888 Live EMCCD camera at 16-bit depth using Nikon NIS Elements AR software. Image cropping was performed in Fiji[71].

## Statistical analysis

All graphs and statistical analyses were performed using GraphPad Prism Version 9.4.1. For all analyses, we used a two-tailed unpaired Student's $t$-test to compare the difference between the two groups. $P$-values are reported in the figure legends.

## Reporting summary

Further information on research design is available in the Nature Portfolio Reporting Summary linked to this article.

## Data availability

Our SBF-SEM datasets are publicly available in the EMPIAR database: EMPIAR-11683. Live-cell imaging and light microscopy data are available in Zenodo: https://doi.org/10.5281/zenodo.10009391. Source data are provided with this paper.

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

## Acknowledgements

We thank Jason (Xiangxi) Liang and Michael Cammer from the NYU microscopy core for assistance with the preparation of EM samples and discussion of light microscopy experiments, respectively. We thank Emily Troemel, Spencer Gang, Kacie McCarty, Joseph Sudar, Mahrukh Usmani, Juliana Ilmain, Nicolas Coudray and Frederick Rubino for critical reading and feedback on the manuscript and all members of the Bhabha/Ekiert labs for helpful discussions. We gratefully acknowledge the following funding sources: SSP-2018-2737 (Searle Scholars Program, to G.B.), R01AI147131 (NIAID, to G.B.), Irma T. Hirschl Career Scientist Award (to G.B.). G.B. is a Pew Scholar in the Biomedical Sciences,

supported by The Pew Charitable Trusts (PEW-00033055). The NYU Microscopy Core is partially supported by NYU Cancer Center Support Grant NIH/NCI P30CA016087, and Zeiss Gemini 300 SEM with 3View was purchased with the support of NIH S10 ODO019974.

## Author contributions

N.V.A., G.B. and D.C.E. were responsible for project conceptualization, experimental design, data analysis and manuscript writing. N.V.A., J.S., C.P. and F.L. performed SBF-SEM sample preparation and data acquisition. N.V.A., C.L. and A.D. performed SBF-SEM analysis. N.V.A. performed optical microscopy experiments and data analysis. M.R. and J.I. contributed to the analysis of parasite development and created data-based animations. G.B. and D.C.E. supervised the project. All authors edited the manuscript.

## Competing interests

The authors declare no competing interests.
