## [Peer Review File · Nature Communications]

3D reconstructions of parasite development and the intracellular niche of the microsporidian pathogen *E. intestinalis*REVIEWER COMMENTS

Reviewer #1 (Remarks to the Author):

This is a very nice paper that provides interesting data on the developmental organization of *E. intestinalis* that is likely applicable to other microsporidia. The authors have done an excellent job of using SBF-SEM for this study and the number of each life cycle stage examined is sufficient to provide a robust data set for comparisons.

Minor comments:

I am not sure that a "harpoon" is the right description for the process of polar tube interaction with its target host cell; however, this is an interesting alliteration and certainly provides a nice image of this process.

Please provide more description of the nucleation center. What is this structure? Does it have any relationship to the Golgi that form during development (but are poorly seen in mature spores)? If not, do you have any speculation on what this structure is and if it is present in mature spores or only forms during development.

As to the Golgi, do your images provide any information on this structure and its development during the *Ei* life cycle?

Did your images also provide any data on polaroplast development?

E. intestinalis is unique among the Encephalitozoon for having a dense matrix in between the organisms in the parasitophorous vacuole (PV); hence its initial description as *Septata intestinalis*. Your images do not seem to show this matrix. Is this what is meant by the comments on vesicular material in the PV? Image 2D does seem to show this matrix material.

Reviewer #2 (Remarks to the Author):

This interesting report uses volume electron microscopy to document microsporidia development and host parasite interactions. The study is well performed and works hard to counter the "low sample number" challenges of the method. The polar tube and mitochondrial observations/quantification and their fine details are particularly interesting, although they remain largely descriptive without in-depth molecular characterisations. My feeling is that the data would be well suited to a specialist journal and would need to provide more substantive mechanistic insights before it would be suited to publication in *Nature Comm.*

Reviewer #3 (Remarks to the Author):

Reviewer comments - Minor Revision

1. P3, first line – including protozoan hosts.
2. P4, first 15 lines of results – these are methods and should be moved to the appropriate section.
3. Figure 1 – please represent your graphs as boxplots to accurately display the data, not bar charts. Please keep individual points on the graph, this provides valuable detail.
4. Figure 3 – please represent your graphs as boxplots to accurately display the data, not bar charts. Please keep individual points on the graph, this provides valuable detail.
5. Figure 4 – please represent your graphs as boxplots to accurately display the data, not bar charts. Please keep individual points on the graph, this provides valuable detail.

6. Figure 6F – please represent your graphs as boxplots to accurately display the data, not bar charts. Please keep individual points on the graph, this provides valuable detail.
7. Figure 6D – please remove the stacked bars and present these next to one another in your early, mid and later groups. Please maintain the colour, but present as box plots.
8. All structural notes are clear and well presented. Please bear in mind that some of the features you present can be referred to as separate names – for example, 'sporophorous vesicle' vs 'parasitophorous vacuole'.
9. Strong discussion and methods.

Overall, a fantastic contribution to the literature and reveals a plethora of novelties on 3D presentation as well as pathological effects.

Final notes:

- please amend Suppl. Table 2, which is cut off.
- Fantastic and informative video clips.

We are grateful for the comments from all three reviewers, which have helped us to improve our manuscript. Referee comments are in **black** and our responses are in **blue**. With the exception of figure references, in the revised manuscript (main text and methods), changes are in **blue**.

Reviewer #1 (Remarks to the Author):

This is a very nice paper that provides interesting data on the developmental organization of *E. intestinalis* that is likely applicable to other microsporidia. The authors have done an excellent job or using SBF-SEM for this study and the number of each life cycle stage examined is sufficient to provide a robust data set for comparisons.

Minor comments:

I am not sure that a "harpoon" is the right description for the process of polar tube interaction with its target host cell; however, this is an interesting alliteration and certainly provides a nice image of this process.

We agree with the reviewer that at a mechanistic level "harpoon" may not, in fact, turn out to be accurate. However, as the reviewer notes, it provides an intuitive visual of the process, which we feel is important and helpful to the reader, so we have chosen to leave it in the manuscript.

Please provide more description of the nucleation center. What is this structure? Does it have any relationship to the Golgi that form during development (but are poorly seen in mature spores)? If not, do you have any speculation on what this structure is and if it is present in mature spores or only forms during development.

We thank the reviewer for this question. Indeed we are very curious as to the structure and composition of the nucleation center, though in our current SBF-SEM images the nucleation center appears as an amorphous electron dense structure associated with the polar tube during development. We do not resolve any clearly defined membrane stacks, vesicles, etc. within the nucleation center that would suggest it is Golgi-like, though that may reflect a technical limitation of SBF-SEM to resolve such small, closely spaced features. For example, we can clearly resolve mitochondrial cristae and the two membranes of the nuclear envelope in many places in the host cell; but, in contrast, we don't reliably resolve the two membranes of the *E. intestinalis* nuclear envelope (which are more closely spaced than those of the host), or the individual membrane layers of the polaroplast. The nucleation center may be morphologically similar to a structure previously called "sac polaire" (Vinckier, 1975) by TEM, which does appear to be associated with Golgi. Both the nucleation center and the "sac polaire" (in some images) have a shape reminiscent of the anchoring disc. We have now made note of the resemblance to "sac polaire" (Page 6) and discussed the possible relationship between the nucleation center and Golgi (Page 8). As to the presence of the nucleation center in mature spores: as noted above, the shape is very similar to the anchoring disc in mature spores, but the texture of the density is not. One possibility is that

the nucleation center morphs into the anchoring disc, but currently we do not have data that directly support this hypothesis. We have now speculated on this possibility (Page 7).

As to the Golgi, do your images provide any information on this structure and its development during the *Ei* life cycle?

The resolution of our SBF-SEM data is somewhat lower than TEM data, and does not allow us to confidently identify the Golgi (see response to previous comment). For this reason, we have not been able to analyze its structure and development during the *E. intestinalis* life cycle.

Did your images also provide any data on polaroplast development?

As with the Golgi, due to the resolution of SBF-SEM data inherently being slightly lower than TEM data, we are not able to identify much detail in the structure and development of the polaroplast from these datasets. We are only able to identify the point at which we first observe the lamellar polaroplast, which is in maturing spores (Page 6).

E. intestinalis is unique among the Encephalitozoon for having a dense matrix in between the organisms in the parasitophorous vacuole (PV); hence its initial description as *Septata intestinalis*. Your images do not seem to show this matrix. Is this what is meant by the comments on vesicular material in the PV? Image 2D does seem to show this matrix material.

In our dataset we do not observe a dense matrix as was previously shown. Surrounding the parasites, we see largely empty spaces, some amount of irregular, unstructured density, and some vesicles. We do not observe the honeycomb-like, regular structure that was previously observed to compartmentalize individual parasites (e.g., Cali, et al, J Euk Microbio, 1993). The differences could be due to technical differences, such as sample preparation, which is different for TEM Vs. SBF-SEM, or it could be due to a difference in the source of the sample. Since it is quite difficult to comment on the differences without a systematic comparison of samples and sample preparation techniques, we have not drawn any conclusions on this, but we have now noted this difference (Page 5).

Reviewer #2 (Remarks to the Author):

This interesting report uses volume electron microscopy to document microsporidia development and host parasite interactions. The study is well performed and works hard to counter the "low sample number" challenges of the method. The polar tube and mitochondrial observations/quantification and their fine details are particularly interesting, although they remain largely descriptive without in depth molecular characterisations. My feeling is that the data would be well suited to a specialist journal and would need to provide more substantive mechanistic insights before it would be suited to publication in Nature Comm.

We thank the reviewer for their review of our manuscript. As there are no scientific points to address, we have made no changes in response to the comment.

Reviewer #3 (Remarks to the Author):

Reviewer comments - Minor Revision

1. P3, first line – including protozoan hosts.

We have now included a reference for protozoan hosts (Quandt et al.,2017).

2. P4, first 15 lines of results – these are methods and should be moved to the appropriate section.

We have shortened this section. We feel, however, that parts of this text are helpful to include in the main text, as it guides the reader through our rationale for the experiment and results shown in Figure 1C-D.

3. Figure 1 – please represent your graphs as boxplots to accurately display the data, not bar charts. Please keep individual points on the graph, this provides valuable detail.

This was a great suggestion, which we have incorporated in the form of violin plots for several figures. However, for this particular figure, there are not enough data points to use a box blot or violin plot, so we have chosen to leave it as is.

4. Figure 3 – please represent your graphs as boxplots to accurately display the data, not bar charts. Please keep individual points on the graph, this provides valuable detail.

We have now represented data in Fig. 3 as a violin plot.

5. Figure 4 – please represent your graphs as boxplots to accurately display the data, not bar charts. Please keep individual points on the graph, this provides valuable detail.

We have now represented data in Fig. 4 as violin plots.

6. Figure 6F – please represent your graphs as boxplots to accurately display the data, not bar charts. Please keep individual points on the graph, this provides valuable detail.

We have now represented data in Fig. 6F as violin plots.

7. Figure 6D – please remove the stacked bars and present these next to one another in your early, mid and later groups. Please maintain the colour, but present as box plots.

As requested, we generated box plots for this figure. However, we feel that the representation with box plots is less clear than the original figure, though both are accurate. We have pasted

below a side-by-side comparison of these representations. If the reviewer feels it is important, we can also include the box plots in the supplement, but we prefer for this not to be the primary representation in the main figure.

8. All structural notes are clear and well presented. Please bear in mind that some of the features you present can be referred to as separate names – for example, ‘sporophorous vesicle’ vs ‘parasitophorous vacuole’.

Thank you for pointing this out. We agree that the organelles are sometimes referred to by an alternate name in the literature, and we now mention that the Parasitophorous vacuole is also referred to as the sporophorous vesicle when we introduce it (Page 3).

9. Strong discussion and methods.

We thank the reviewer for their positive comments.

Overall, a fantastic contribution to the literature and reveals a plethora of novelties on 3D presentation as well as pathological effects.

Final notes:

- please amend Suppl. Table 2, which is cut off.

Thank you for pointing this out. We have amended Supplementary table 1 so that it is no longer cut off.

- Fantastic and informative video clips.

Thank you!

REVIEWERS' COMMENTS

Reviewer #1 (Remarks to the Author):

This is an excellent paper that adds significantly to the literature and our understanding of these enigmatic pathogens.

The authors have addressed all of my previous questions.

I agree with the authors that Figure 6D was clear in the original manuscript and that the box plot version suggested by reviewer 3 does not present the information as clearly as the original figure.

We are grateful for the comments from all reviewers, which have helped us to improve our manuscript. Referee comments are in **black** and our responses are in **blue**. With the exception of figure references, in the revised manuscript (main text and methods), changes are in **blue**.

Reviewer #1 (Remarks to the Author):

This is an excellent paper that adds significantly to the literature and our understanding of these enigmatic pathogens.

The authors have addressed all of my previous questions.

I agree with the authors that Figure 6D was clear in the original manuscript and that the box plot version suggested by reviewer 3 does not present the information as clearly as the original figure.

We thank the reviewer for their positive comments.